# Behind the smile: qualitative study of caregivers' anguish and management responses while caring for someone living with heart failure

Jennifer Wingham,[1,2] Julia Frost,[3] Nicky Britten[3]

► Prepublication history and additional material are available. To view these files please visit the journal online (http://dx.doi.org/10.1136/bmjopen-2016-014126).

[1]Research, Development and Innovation, Knowledge Spa, Royal Cornwall NHS Hospitals Trust, Truro, Cornwall, UK
[2]Primary Care Research Group, Institute of Health Research, University of Exeter Medical School, University of Exeter, Exeter, Devon, UK
[3]Third Gap Group, Institute of Health Research, University of Exeter Medical School, University of Exeter, Exeter, Devon, UK

**Correspondence to**
Dr Jennifer Wingham;
jennywingham@nhs.net

## ABSTRACT

**Background** Caregivers support self-management in heart failure but often experience stress, anxiety and ill health as a result of providing care.
**Aims** 1. To identify the factors that contribute to the experience of anguish. 2. To understand how caregivers learn to live with what is frequently a challenging and demanding role.
**Methods** Individual interviews with caregivers who had been caring for someone with heart failure for a minimum of 6 months. We used thematic analysis to inductively analyse transcripts.
**Results** Twenty-two caregivers, from three centres in the United Kingdom, took part in individual interviews. The caregivers were aged between 39 and 84 years, and six were men. Twenty were in spousal or partner relationships. We found that caregivers often hide the extent of their emotional stress or anguish. We identified four main themes with explanatory subthemes—emotional impact (fear for the future and sense of hopelessness), role definition (changing sense of who I am, reduced resilience, learning care skills, role conflict and changing role), exclusion (exclusion by the cared-for person and by health professionals and feeling alone) and ignoring one's own health—that were associated with anguish. From these findings, we produced a caregiver needs assessment model in the context of caring for a person with heart failure.
**Conclusions and implications for practice** Caregivers have many unmet and hidden needs. Primary care health professionals are well placed to meet the needs of caregivers. The model may be used by health and social care professionals to identify needs and to provide caregivers with targeted practical and emotional support; and for researchers developing interventions to enhance self-management in heart failure.

### Strengths and limitations of this study

► We provide evidence to understand the enduring emotional impact of informal caregiving
► We present an evidence informed model for health professionals to assess the emotional and psychological impact of caring
► Caregivers were located in three centres in England
► This is a further analysis of a data set
► The study is focused on a single long-term condition

### INTRODUCTION

There are over 100 million unpaid caregivers in Europe, but as this is a self-reported figure, it is likely to be an underestimate.[1] In the UK alone, their contribution is worth £132 (€173) billion a year to the economy.[2]

Caregivers have an important role in supporting chronic disease management, although this often comes at considerable personal cost and many are uncertain about how to best provide this support.[3–5] The majority of caregivers have health and well-being problems of their own, either pre-existing or arising from the burden of their caregiving activities.[2 6] Caregiver burden is understood to be due to the tasks of caring (objective burden) and the distress associated with the role (subjective burden).[7] The objective burden is due to activities that include physical tasks such as washing, dressing, feeding and assistance with mobility, sometimes also performed at night. Other caregiving activities support self-management of chronic conditions such as heart failure, including assistance with identification of signs and symptoms of deterioration, taking action during an emergency, assisting with blood pressure monitoring and medication management. It also includes supporting adherence to dietary restrictions, providing emotional support, supporting exercise and physical activity, providing transport and maintaining safety and liaison with health professionals.[8–10]

Heart failure is a long-term condition characterised by breathlessness, fatigue, signs of fluid retention, loss of appetite and short-term memory loss.[11] Those who experience symptoms have an increased risk of anxiety and depression as they live with uncertainty about the condition and often

experience social isolation.[11–13] The condition requires skilled self-management including coping with a complex medication regimen, weight monitoring for fluid control, dietary restriction, exercise and physical activity and management of well-being.[14] With an ageing population, the number of people with heart failure is expected to grow,[11 15] and this will have an impact on family members and informal caregivers who have an important role in assisting self-management, including exercise adherence.[16] There are, however, currently few examples of effective self-management interventions that include caregivers.[16 17] Al-Janabi *et al* in a recent BMJ editorial recommended that caregivers be included in healthcare decisions, and more data need to be collected on the impact of patient interventions on caregivers.[18]

This paper is a further analysis of an existing data set collected in a qualitative interview study that identified the needs of caregivers.[9] The study sought to identify the needs of caregivers to inform the development of a home-based facilitated cardiac rehabilitation programme: Rehabilitation Enablement in Chronic Heart Failure (REACH-HF).[19] During the study, the researchers noticed that when knocking at the door of respondents' homes, caregivers often opened the door with a smile and welcome, giving the appearance that they were coping. Once the interview commenced and rapport was established, a different impression often emerged as many caregivers cried, were angry or appeared emotionally flat, with few expressing satisfaction with their role. In team discussions, we used the term anguish to describe the extreme responses of some of these caregivers. Anguish is defined as 'mental suffering which includes fright, feelings of distress, anxiety, depression, grief and/or psychosomatic physical symptoms'.[20]

Our aim was to identify the factors that contribute to the experience of anguish and understand how caregivers learn to live with what is frequently a challenging and demanding role.[9] The purpose of this paper is to inform healthcare providers and researchers about the often hidden perspective of caregivers and to inform practice and the development of cardiac interventions that include caregivers.

## METHODS

The study's methods, based on qualitative interviews, have been reported in full elsewhere.[9] Our intention was to understand the perspectives and lived experiences of caregivers. In our first paper, we used thematic analysis to provide a rich description of the entire data set as described by Braun and Clarke.[21] Thematic analysis can also be used to provide a more detailed inductive account of a particular theme. In our first paper, we reported that caregivers needed to manage mental health, well-being and sleep. We now report a more detailed analysis of those caregivers in our study who were struggling to manage their mental health, well-being and sleep and who were anguished. Thematic analysis allowed us to

contextualise the ways individuals give meaning to their experiences and reality.[21] One of the interviewers is an experienced nurse and qualitative researcher (JW). All authors have PhDs in social science or medical research and are female. The second interviewer is male and has an MPhil in geography. Participants were not previously known to the researchers.

## Recruitment

We purposively sampled caregivers from three centres across England: Cornwall, Birmingham and Leicester. Caregivers were defined as 'anyone who cares, unpaid, for a friend or family member who due to illness, disability, a mental health problem or an addiction cannot cope without their support'.[10] The caregivers had at least 6 months' experience as caregivers. Potential caregiver participants were approached by clinical cardiac health professionals. The caregivers were given a letter inviting them to participate in the study, a patient information sheet, a questionnaire collecting demographic and socioeconomic information and a stamped addressed envelope. The invitation letter included a tear-off slip to allow caregivers to provide contact details if they wished to be contacted by the study team. The final sample selected from the questionnaires represented a range of demographic and social factors including ethnic diversity, household income, rural and urban dwellers, relationship to the cared-for person, length of time as a caregiver and level of education.

## Interviews

Using a topic guide and following informed consent procedures, two researchers conducted the interviews (JW and DT). Topics included: reaction to the diagnosis of heart failure, what formal support services were used and what was missing, caring for one's own health and well-being, social support structures and recommendations for new caregivers (see Box 1). During the interview, the researchers used recognised qualitative techniques such as reflecting back to the caregiver what has been said. Field notes captured contextual information such as the general location, observed relationship with the cared-for person and any interruptions.[22 23]

## Analysis

All audio-recordings were transcribed verbatim by an experienced transcriber. JW and DT ensured that the caregivers' transcripts were anonymised and checked for accuracy. Nvivo 10 was used to manage data as thematic analysis was conducted.[21] This is a six-step process that involves familiarisation with the data, identifying initial codes, identifying initial themes, reviewing and revising the themes, naming the themes and assigning descriptions of the themes and finally producing the report. For this paper, JW and JF focused on initial themes, reported in our earlier paper,[9] and associated with any negative impact arising from being a caregiver. From this, we constructed a model as a visual technique

**Box 1    Topic guide**

Tell me about an average day looking after the person you provide care for?
Prompts: Specific tasks, for example, taking medication, adjusting doses, encouraging daily weights, monitoring your loved one's state of health, managing appointments, feelings through the day
How did your loved one react and respond to having heart failure?
Prompts: Does the person with heart failure talk about their condition, do they avoid talking about the condition, expect the caregiver to take over or manage their condition well?
How did you react to your family member having heart failure?
Prompts: Feelings, desire to care, feeling burden, feeling confused, not wanting to worry other family members about own needs, what helped to adjust?
What does caregiving involve? Has this changed over the time you have been a caregiver?
Prompts: Look for talk that may indicate any reaction that may indicate a style of management: positive proactive person, sense of duty, enjoys care giving, likes control, feels powerless. Look for changes over time and strategies to manage the condition either as an individual or together with the person who has heart failure. How do they define their role?
Tell me about any discussions you may have had with your loved one about how you would manage the heart failure?
Prompts: How did the discussion take place? Who led the discussion? Look for similarities and differences between the cared for and the caregiver.
What support have you received in your role as a caregiver?
Prompts: Formal support from health and social services—from the doctor, hospital, specialist service, online services—healthtalkonline or heartmatters.org or other. Informal support—friends, neighbours, family, support groups, online blogs, forums.
Tell me about any specific information or advice about caring for someone with heart failure you may have received?
Prompts: What was most helpful? What was least helpful? What would you have liked? Do you feel you have an adequate understanding of your 'loved one's' condition and what you can do to help?
Tell me what you do if anything to look after yourself?
Prompts: Actions to manage stress, for example, look for signs of smoking, alcohol, exercise, respite care, sitting service, going out with friends shopping, bath, among others.
What advice would you give to a new caregiver of someone with heart failure?
What other information or support would you like to have had at the start or would like to have now?
Prompts: Potential topics for the manual or for the nurse facilitator.
A nurse will work with the individual with heart failure and their caregiver as they use the manual. Is there anything you would like to say about how the nurse should work with you?
Prompts: Face-to-face contact, telephone, same time as the person with heart failure or separately.
Summarise the interview
Is there anything else you want to add?

to depict the complexity of anguish and how the themes are interrelated (see figure 1).

## ETHICS

Ethical approval was granted by National Research Ethics Service Committee South Central-Southampton B (12/SC/0643). All participants gave informed consent.

The Royal Cornwall Hospitals National Health Service (NHS) Trust acted as sponsors of the study and is responsible for oversight of the research governance arrangements.

## PATIENT AND PUBLIC INVOLVEMENT

The REACH-HF programme of research has an established and embedded Patient and Public Involvement (PPI) group of nine patients and caregivers led by a lay chair.[24] The research questions and topic guide were developed with the group through meetings, email and telephone consultations. Both patients and caregivers shared their experiences of how caregiver needs were often not considered as part of healthcare consultations. The PPI group gave advice on how to ask sensitive questions. The recruitment process was codesigned with PPI

members. The lay chair commented on the feedback sheets that were sent out by post. These were sent after the lead researcher contacted each participant by telephone to ask if they wanted to receive the results.

## RESULTS

Twenty-two caregivers took part in individual interviews between January and December 2013. All interviews took place in the caregiver's home, except for one which, at the request of the caregiver, took place in the research unit. The audio recorded individual interviews ranged between 42 and 87 min with a mean of 62 min. The caregivers were aged between 39 and 84 years, and six were men. Twenty caregivers were in spousal or partner relationships, one was a father/son relationship and the other was a mother/daughter relationship. Eighteen described themselves as White British, one as Black British, one Black Caribbean and one Indian. Three were employed, 1 was a full-time working-age caregiver and 18 were retired. Twelve caregivers reported some physical or mental health illness. Four participants had a household income of less than £10 000, 11 participants between £10 000 and £20 000 and 5 participants between £21 000

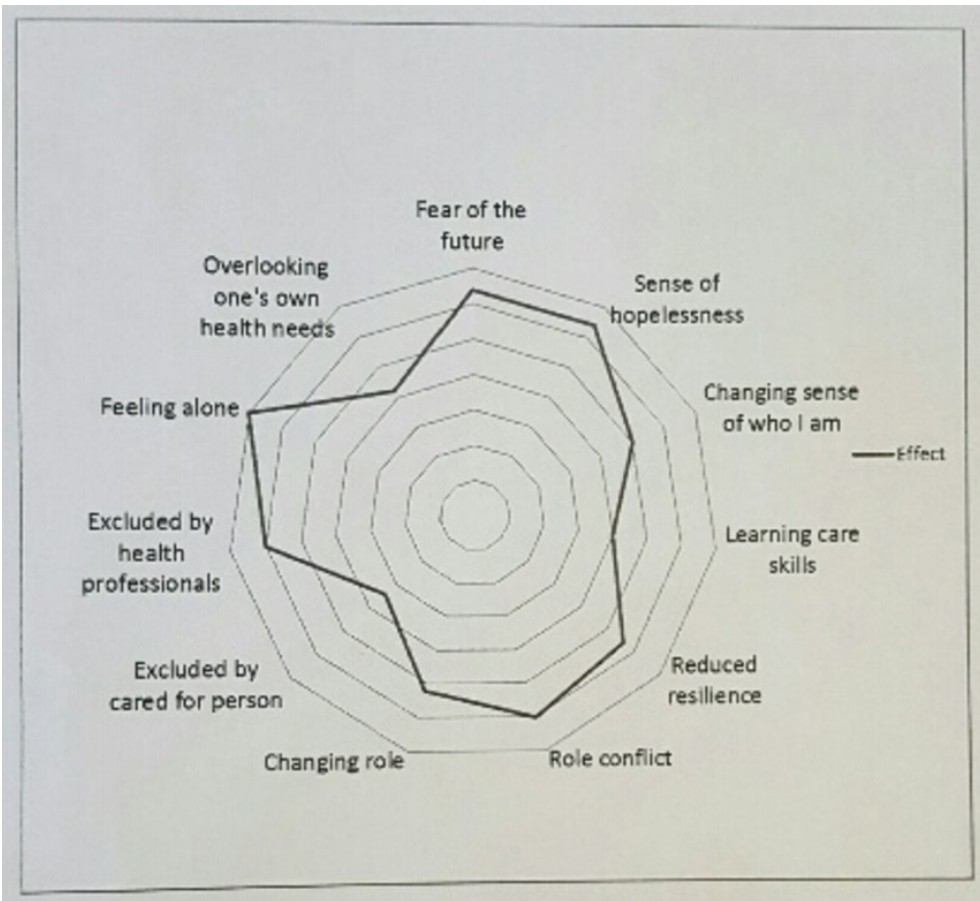

**Figure 1** Caregiver Needs Assessment Model. The factors should be identified and mapped (by the caregiver alone or with their health professional). The factor with the lowest importance is charted nearest to the centre of the diagram with the factor of the greatest importance nearer the outside.

and £30 000. One had an income over £31 000, and one withheld financial information. Six were educated to degree level or above. Twelve caregivers requested that the cared-for person was present.

When we identified the needs of caregivers, we noted that there were considerable data about the causes of stress and difficulty adjusting to the emotional and psychological aspects of being a caregiver.[9] In this analysis, we identified four overarching themes (emotional impact, role definition, exclusion and ignoring one's own health needs) to develop the Caregiver Needs Assment Model as experienced by caregivers in this study.

## EMOTIONAL IMPACT
The development of enduring anguish often began with an emotional response to witnessing a partner, child or parent becoming critically unwell. This included shock, fear, upset and distress. Several of the caregivers had sat for several days in critical care environments, by the bedside of someone seriously ill, hoping for a recovery. Even when a 'good' recovery had been achieved, several caregivers spoke of enduring fear of the future.

### Fear of the future
Those who were anguished struggled to cope as there was no clear predictive illness trajectory. They were aware that death in heart failure may be sudden or occur after several years of considerable ill health. Caregivers were fearful of a future being left alone, which was often considered to be worse than the current experience or were grieving for the loss of the planned future.

This caregiver had previously coped with many life challenges including her own poor health; however, she now felt uncertain that her husband would live and how she would cope.

"I mean, we've had very bad ups and downs and things like that, and a shortage of money and all but we've always come through it. And this time I didn't think we were going to and it was all that everything I felt was going to burst. I felt sometimes that my head was going to burst, I was having bad heads and all this, and it's nothing but pressure." C2

### Sense of hopelessness
The sense of hopelessness was more likely to occur when the caregiver understood that there is no cure for heart

failure. When the person with heart failure slept, some caregivers were particularly concerned that the person may die. Times when the caregiver was alone were also challenging, as they wrestled with their thoughts; this was particularly noticeable for those with little social support.

> But you've got the anxiety of… sorry [C is crying]. What are you going to find if you do go up there? [C's voice breaking up]. C13

## ROLE DEFINITION
### Changing sense of who I am
These caregivers indicated that becoming a caregiver meant adapting from life as known (and sense of who they are) to a new life (and sense of who they are now or will be in the future). Those experiencing enduring anguish had great difficulty making the transition and struggled to create or retain a sense of self/identity. This was also associated with fear of the future as caregivers recognised they will require further transition to life as an 'ex-carer' in the future.

> I have felt down. I've said, Why me? Our lifestyle was going so well and then all of a sudden… you've got this bombshell and you're just expected to just change. C24

### Learning care skills
The caregivers who were not coping had an overwhelming feeling that they lacked the skills to support a person with heart failure. Typically, caregivers were concerned with a complex and ever-changing medication regime. Some community cardiac nurses worked closely with a few of the caregivers to provide education and support around medication and management systems, and their support ameliorated the caregivers' anguish. These caregivers demonstrated development of care skills.

> I found ****** [cardiac nurse] particularly helpful because I came in blind and it took me a long time to realise I was a carer. You learn gradually and I learnt from her more than anybody else how to be a pharmacist, a carer, I'm the taxi driver, a range of jobs. It's like the tablets and getting the medication right; it took me a long time to realise that when it comes to water control it's down to me to decide whether I'm going to give him any extra or not. C5

### Reduced resilience
Many caregivers described their role as being '24/7,' indicating there was no break and the relentlessness of caregiving had eroded their sense of resilience. Some provided physical care through the day and night, while others were constantly watchful or hypervigilant for signs of heart failure. This was particularly evident where the person they were caring for had experienced more than one acute illness and faced the possibility of death.

A common coping mechanism for those who had overcome anguish and developed resilience was to have respite away from the caregiving environment even if it was for a limited period of time.

> I potter or I might go to [name of supermarket] or something and just get out, because I get to the point, I think, 'If I don't get out of this house, I'm going to scream!' C4

### Role conflict
Many caregivers described the difficulty of juggling their other roles or household activities with the duties of supporting a person with an unpredictable chronic illness. This lead to a sense of guilt and inadequacy as they felt these roles were at times unfulfilled.

> That's hard. Especially when you, and two children, 'Mum, can we do this?' 'No, sweetie, because I can't, not really, because your dad's not well enough' and… that and the guilt with the children is hard. C4

### Changing role
The caregivers spoke at length about their sense of responsibility and the challenges faced, as their day was often difficult to plan. They were forced into a 'go with the flow' or 'living in the moment response' as they responded to situations as they unfolded rather than planning ahead. The dynamic nature of the role meant there were times when the caregiver needed to discern when to step in and take control or leave the locus of control with the individual.

> There isn't an average day. You take each day as it comes. Um, no two days are the same. C12

## EXCLUSION
Caregivers who were anguished were likely to feel excluded from decisions affecting care.

### Exclusion by the cared-for person
Those who experienced anguish reported times when the cared-for person excluded them from consultations or declined to talk about their situation, management plans or plans for the future. This could be due to the quality of the pre-existing relationship or because the cared-for person tried to protect the caregiver. This raised anxiety about the future and lead to hypervigilance. This caregiver perceived the interview as a therapeutic intervention and relished the opportunity to talk to someone about how she felt.

> …This is why I said I'd do it [take part in research], because I would sometimes need to talk to somebody. Because I'll talk to him and he'll change the subject

or something and I think, 'Well, it's not getting nowhere, is it?' C17

### Exclusion by health professionals

Exclusion from health consultations added to the sense of anguish as the caregivers were left fearing the worst and had to make decisions from ill-informed positions. Some caregivers also indicated that even when they were present in consultations, their own needs were often overlooked. Being excluded from consultations could lead to a tendency for the caregiver to be overprotective of the cared-for person or experience psychological distress and anguish.

> The ladies [health professionals] here, you're listening to them, but they're not talking to you. They're talking to the patient. C15

### Feeling alone

Caregivers in this sample who felt anguished were usually those who have little social support. Some described how they became isolated over time, particularly in areas where there were a high proportion of elderly people or where the household had recently relocated to a new community. Sometimes the isolation was because the caregivers present themselves as coping and were reluctant to ask for help. We noted that those who were describing anguish were often protecting grown-up children from the extent of their emotional pain and their struggle to cope. Others also had difficulty accessing carer support groups.

> And also you feel that you've got to put a brave face on it. 'I'm alright, I'm alright'. Until that time that you've got to admit to yourself, 'No, I'm not alright', and need to admit, especially, to our [children] I suppose. C18

In contrast, caregivers who adjusted to their role were those who were in good health and had a good social support structure.

> I keep reasonably good health and I feel that I can cope with it… We have a fantastic family….I get tremendous support. C6

### IGNORING ONE'S OWN HEALTH NEEDS

Many caregivers in this study had significant health challenges, including bone and joint conditions, cancer or depression and anxiety. These factors also contributed to the emotional impact of heart failure and sense of anguish in their lives and the experience of role conflict, as the caregiver could not always provide the care they would like to provide. They often put the health of the cared-for person before their own and frequently delayed going to the general practitioner (GP) until they themselves were in crisis. There were examples of delaying hip replacement surgery (C2), unreported chest pain (C1) or mental health problems (C4, C17, C18, C22 and C26).

This was more likely to occur when the cared-for person was severely affected by heart failure or other conditions or when there was a lack of a supportive social network.

> I think it did all come to a little bit of a head, a couple of months ago, and I just had enough and I went to my own GP, they just said I needed to take some time off work. So I was off work for a month. C18

Two caregivers had previously been the cared-for persons themselves and had to make considerable psychological adjustments. This highlights that caregivers are sometimes the one who is least ill in a relationship.

### DISCUSSION

The existence of caregiver-related stress and burden and the consequent ill health have been recognised for many years. We present the first evidence of informed understanding of enduring anguish in caregivers that we are aware of. The Care Act (2014)[25] and the new NHS England's toolkit[26] for commissioners set out requirements for caregivers to be identified and for care plans involving both healthcare and local authorities, but there are few practical tools for identifying needs and providing care.[26] Our model (figure 1) indicates that some of the themes are related as we found that those who were particularly anguished experienced coexisting subthemes. We note that our model has similarities to the Lifepsycholo QOL model for patients living with chronic illness.[27] Expressed as a circular diagram, our model elaborates on theirs and enables factors to be identified and mapped (by the caregiver and/or their health professional), from caregiver themes with the lowest importance in the middle of the diagram to those with the greatest importance nearer the outside. In this way, needs can be openly discussed to aid identification and prioritised enabling targeted support to be planned and provided. Change over time can then be reported on further diagrams, for example, when an intervention is provided and/or when symptoms or circumstances change.

### Emotional impact

We provide evidence for a model for understanding the phenomenon and the consequences of this often-hidden emotion. This addresses the recommendation from Montgomery and Kosloski[28] for research about the specific causes of caregiver distress. We found that the presence of anguish was often only revealed through skilful interviewing and rapport building. At the beginning of the interviews, caregivers tended to talk about the challenges faced by the person with heart failure and their treatment plans. While this gave us useful contextual data, the researchers often had to reorientate the interview to focus on the experience of the caregivers and their unmet needs. Possible explanations for this phenomenon are 'normalisation' or 'passing.' Normalisation refers to a process where the individual treats an illness or disability as routine,[29] often in order to avoid stigmatisation.

Caregivers present 'a face' as coping as they believe they are expected to cope, and it is this phenomenon that health professionals need to be aware of as they tailor care to the family. As part of the UK's Care Act 2014,[25] caregivers now have the right to have their needs assessed and to be involved in care planning (including hospital discharge planning) unless the person they are caring for objects. The Caregiver Needs AssessmentModel (figure 1) may be used as a tool by health and social care professionals, including doctors and community-based cardiac nurses, as an assessment tool as they care for caregivers to enable them to go 'behind the smile' presented.

Anguish exists where there is a loss of hope and where the caregivers realise that there is a difference between their expectations of life and the reality in which they exist. Anguish is often associated with grief as an enduring emotion. Price[30] suggests that anguish is not solely confined to grief but is an emotion experienced because of loss, including separation, such as being apart from friends and family. Caregivers in our study acknowledged their loss of their previous quality of life, and they either made explicit reference or alluded to the prospect of death to come. The literature suggests that there are positive aspects to anguish, as it enables an individual to reflect on their situation and seek a sense of 'being' to discover personal ability, talent, strength and courage.[31 32] This enables caregivers to find a resolution through altruism or fulfilment in successfully providing care and managing their role.[33] We have demonstrated that some caregivers are able to work through their anguish and find ways of managing their situation and emotions while acknowledging that, ultimately, they will face the prospect of watching someone they love or care for die. Similarly, Jowsey et al[34] in a recent qualitative study identified the significant impact of worry on caregivers and that there are both positive and negative impacts as it informs their sense of self, motivation and view of what the future may entail. Worry may be seen as the temporal process that engages caregivers' time, while anguish is the long-term emotional outcome.

### Role definition

There has been much work on the impact of chronic illness on those people living long-term health conditions.[34] Bury identified the profound impact on personhood of what he called 'biographical disruption.'[35] This disruption may be in terms of consequences including symptom management and the significance of the condition, in particular dependency and requiring help.[36] There is therefore a similarity with caregivers who are experiencing disruption to their lives and are adapting to life as a provider of support.

We were struck by the considerable amount of time that many of these caregivers were spending on their role as caregivers and in adapting to their role. Charmaz and Paterniti[37] recognised that the problems chronic illness present 'fall squarely' on ill people and their families.[37] Finding solutions to these problems may be described

in terms of 'work' that encompass practical, emotional and biographical management. Work therefore includes practical household management, management of the condition and emotional consequences, personal expectation of the future and negotiation of help.[38 39] For caregivers, this involves significant identity reconstruction and learning of new skills.

The caregivers in this study who struggled to cope were typically those who realised that they were required to learn new skills often associated with the identity of professionals. In particular, management of medication was a common cause of stress particularly in the early days and weeks following hospital discharge. Those who coped best had a partnership relationship with the cared-for person and/or significant support from community cardiac nurses as they made the transition to becoming caregivers. This finding supports Montgomery and Kosloskis' Caregiver Identity Theory,[28] which 'acknowledges caregiving as a journey that includes a series of transitions that result from changes in the caregiving context and in social norms that are grounded in familial roles and culture.'[40] Our research demonstrates that many caregivers do not cope with the complex transitions required as they construct or revise their sense of identity.

Anguished caregivers experienced chronic fatigue and a loss of resilience. Resilience described by Marsh and Johnson 'is the ability to rebound from adversity and prevail over the circumstances of one's life.'[41] This involves the family reacting to stress, managing the effect of the stress and restoring or adapting the family as a unit. The reduced resilience seen in this study was compounded by the need to adapt to a dynamic role, meaning that they have to adapt by 'going with the flow' or by 'living in the moment.' We saw examples where this was felt to be the best strategy, while others struggled with the loss of personal control.

### Exclusion

Heart failure is unpredictable and has an uncertain trajectory. Many people living with heart failure work in partnership with caregivers to manage the condition.[42 43] The routine experience reported for many of the caregivers was that, in healthcare, they are seldom asked about their own feelings and needs and experienced superficial relationships with healthcare professionals who are focused on the person with the condition. We have, however, been given examples where some cardiac nurses have worked closely with caregivers as they jointly provided support for the cared-for person, but this was not the experience of all the caregivers.

Exclusion also encompasses social isolation and occurs where there are difficulties with limited social networks, the web of social relationships including formal and informal networks.[44 45] We were struck by the lack of awareness of support groups or the availability of carer assessments.

> **Box 2   Potential questions for health professionals to use with caregivers**
>
> ► What does heart failure mean to you?
> ► Is there anything you would like to know about?
> ► It is common for some family members worry about what they should do. What worries you?
> ► What do you do to look after your own health?
> ► What would you like help with?
> ► Do you have any family members or friends you can talk to or ask for help?
> ► What do you do for relaxation or just for you?
> ► What do you think about the future?

### Ignoring one's own health

This Caregiver Needs Assement Model adds to our understanding of the health and social consequences caregivers experience when their needs are not being met.[9] Many of these caregivers expressed appreciation of the opportunity to talk to the interviewer, implying a desire for a therapeutic effect. They often said that no one had ever asked how they were feeling or what their needs were. One caregiver told how she found the participant information leaflet helpful as it contained a list of support organisations. Many of our caregivers were unaware of what services may be available and only two had received a carer assessment. In our previous work, we identified the global needs of caregivers and presented a model by which health and social care professionals may target care. It demonstrated how caregivers need to mobilise their internal and social resources in order to support a person with heart failure. Following our further analysis, we have demonstrated that caregivers face considerable hidden challenges and that there are consequences of living with unmet needs and enduring anguish. We suggest health professionals, especially those in primary care, are well placed to use the model of caregiver anguish to assess the emotional and psychological impact of caring to plan family-centred care dealing with the factors identified where possible. The health professional may need to state that while family or friends who are providing support feel that they should be able to cope, many caregivers struggle. Box 2 provides some suggested questions health professionals may use to get behind the smile. These questions may be adapted to the context in which the health professional works. We further suggest that modern healthcare has focused on measuring and treating stress, anxiety and depression while neglecting the experience of anguish and the emotional consequences on the role, health and well-being of the caregiver. There may also be significant risks as a result of anguish as it may lead to individuals to act in extreme ways to resolve the distress including suicide, violence or murder.[31 46]

### LIMITATIONS OF THE STUDY

We acknowledge that there are limitations to the study. We did not set out to investigate the phenomenon of anguish in caregivers; therefore, there may be other factors that contribute to the presence of anguish that we have not identified. In addition, our understanding of anguish is based on research that focused on a single long-term condition. Despite this, we have presented rich qualitative research and many of the caregivers were also supporting people self-managing other long-term conditions such as arthritis.

### FUTURE RESEARCH

We agree with the recent call for interventions for long-term conditions to include an assessment of the impact on caregivers.[18] This study, together with our research that has identified the needs of caregivers, provides evidence for supporting the development of interventions that include caregivers in heart failure self-management. We suggest here that the model could be tested in caregivers supporting people with other long-term conditions and in palliative care for heart failure.

### IMPLEMENTATIONS FOR PRACTICE

► Caregivers should be actively included in healthcare management decisions.
► Caregivers' emotional experience and sense of burden should be assessed as part of routine and ongoing care and action taken to support them in their role.
► New healthcare interventions or collaborations should include an assessment of impact on caregivers.

**Acknowledgements**  We thank Dave Turner for assisting with the interviews, Sarah Buckingham for transcribing the interviews, the Rehabilitation Enablement in Chronic Heart Failure Patient and Public Involvement group, Katy Oaks Librarian, Royal Cornwall Hospital and not least the caregivers who were candid about their experiences. We acknowledge the REACH-HF investigators led by the Chief Investigators Rod Talyor and Hayes Dalal.

**Contributors**  JW, JF and NB designed the study. JW collected the data. JW and JF undertook analysis. JW wrote the manuscript with contributions from JF and NB.

**Funding**  All authors have completed the International Committee of Medical Journal Editors uniform disclosure form at www.icmje.org/coi_disclosure.pdf and declare: This paper presents independent research funded by the National Institute for Health Research (NIHR) under its Programme Grants for Applied Research scheme (RP-PG-1210-12004). For Professor Nicky Britten, this work was partially funded by the UK NIHR Collaboration for Leadership in Applied Health Research and Care of the South West Peninsula. The views expressed in this publication are those of the author(s) and not necessarily those of the National Health Service, NIHR or the Department of Health.

**Competing interests**  None declared.

**Ethics approval**  NRES Committee South Central-Southampton B (12/SC/0643).

**Provenance and peer review**  Not commissioned; externally peer reviewed.

**Data sharing statement**  No additional data is available.

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
