## [Reviewer comments · BMJ Open]

ARTICLE DETAILS

TITLE (PROVISIONAL)	Behind the Smile: A qualitative study of caregivers' anguish and management responses while caring for someone living with heart failure.
AUTHORS	Wingham, Jennifer; Frost, Julia; Britten, Nicky

VERSION 1 - REVIEW

REVIEWER	Jacqueline Jones PhD RN FAAN University of Colorado College of Nursing, USA
REVIEW RETURNED	25-Oct-2016

GENERAL COMMENTS	This is an interesting paper and a very important topic. However, in its current form the 'qualitative' approach, the underlying epistemology and therefore the grounding of the interview data, its analysis and resulting interpretation is limited or incomplete. The issue could be one of publication expectation as the authors declare qualitative expertise. The type of study design is not provided. What is a qualitative interview study-for the primary study? How were interviews framed? What were the guiding questions and is a guide provided for the reviewer/reader to grasp the boundaries of this emergent secondary analysis? In addition to the paper needing more depth and parameters to the qualitative methodology-analysis alone is insufficient - the paper could be strengthened by re-contextualizing the findings with other caregiver in heart failure experience studies that exist in the discussion.
---

REVIEWER	Kasey R. Boehmer Mayo Clinic, USA
REVIEW RETURNED	17-Nov-2016

GENERAL COMMENTS	Thank you for the opportunity to review this paper. Overall it is a strong qualitative study. I have a few comments that I would consider before publication, particularly related to the results, discussion, and usability of the model. There are also quite a few typos/grammatical errors that I have tried to point out, but I would suggest additional proofing as well. Abstract: Page 2, line 17 and 18: thematically analyzed reads a bit odd. Perhaps rephrase to: "we used thematic analysis to inductively analyze transcripts." Page 2, line 23: 6 should be "six" Page 2, line 33: I do not think that the "poor health" theme is a good description for what you discuss, because it reads like an outcome.
---

What you really describe is more like poor attention to the caregiver's own health.

Background:

Page 3, line 31-41: The sentence that begins with "Other caregiving" is too long and difficult to read. Separate for readability.

Methods:

Page 4, Line 37: Can you please describe in what ways you purposively sampled? Were you seeking distribution among ages and genders? Geography? Income? Something else?

Page 5, Lines 5-8: This should be phrased that authors had PhDs, MPhil, etc. Educated to reads oddly.

Page 5, Line 28: There is a typo in the sentence with analysis thematic analysis. Not sure how it should be revised, but should be attended to.

Results:

Page 6, line 38: Explanatory subthemes to explain is redundant.

Page 7, line 29-30: This quote does not really get at what is described for me. It does not contribute to furthering my understanding of the phenomenon. Can you please expand the quote with more context, or choose a different, more illustrative quote?

Page 9, line 45-46: The idea that the reason they decided to participate in research so that they can talk to someone deserves calling out. It is not uncommon for people to think that talking with an interviewer has a therapeutic effect, and I think deserves attention in the manuscript. Does this illustrate a lack of resources or knowledge of resources for therapy services?

Page 10, line 15: I would rephrase this, as the relationship between caregiving anguish and social support appears causal as currently stated. However, we cannot know that this is a causal relationship from this study.

Page 10, line 44: I think the section on poor health should be renamed (see abstract comment), and I think it also deserves expansion. Compared to the other themes, it seems under-developed.

Discussion

Page 13, lines 8-10: I think this section deserves some citation of Bury, and his concept of biographical disruption, which was documented in patients. Other helpful citations related to these concepts of identity and reformation can be found in the work of Charmaz and Corbin and Strauss.

Page 13, line 26: The subheadings in the discussion are not consistent. Some are bold, some italic, and some both.

Page 13, line 31: focused is misspelled.

Page 19, figure: I think this is a great idea. However, after having done design research in developing discussion and decision aids for clinical encounters, I would suggest in addition to providing the figure, providing a script of how clinicians should use the tool, what questions they should ask. I would not assume that they will know what to do with this, and I think the usability of the tool would be helpful. Also, if it is to be used in collaborations with patients/caregivers and clinicians, I would suggest making the language more friendly to a lay audience. I wouldn't expect a caregiver to know what "proto-professional skills" means, nor would I expect a clinician to know how to properly explain the idea to the patient. After these changes, it wouldn't hurt to put this in the hands

	of some colleagues working with caregivers for them to try out with your instructions. This will bring up problems that you can refine upon.
--	--

VERSION 1 – AUTHOR RESPONSE

Reviewer one

Page 4 and 5. Methods. The reviewer requested more detail about the underlying epistemology of the study. We have added more detail to the methods section on page 4. We have explained how the paper offers an account of this further analysis and the approach used. We realised that the paper implies secondary analysis, we have clarified in the paper that this is further analysis of our existing data set. We have now shown how it relates to the first paper from this data set. We have added much more detail of the methods used as set out in the specific points raised by the second reviewer. The additional text in the recruitment section has provided more detail as requested including how the sample was selected.

What were the guiding questions and is there a guide for the reviewer/reader?

Page 5, more information has been provided about the topics covered in the interview. Table one (page 22) has been added showing the topic guide used. The text relating to the topic guide has been moved to the first time the topic guide was mentioned to improve the flow of the methods section.

Reviewer two

Abstract (Page 2)

We have changed to thematically analysed to read as “we used thematic analysis to inductively analyse transcripts” as suggested.

6 has been changed to six

We have changed the theme ‘Poor health’ to ‘Ignoring one’s own health needs’. This differs from the suggestion of the reviewer but we felt our alternative sounds less value laden and reflects the action of the caregiver. We have added a little more detail to develop the theme on page 12.

Background

We have clarified in the first sentence that we are referring to unpaid caregivers.

Page 3, as suggested we have broken down the sentence “Other caregiving into two sentences to aid readability.

Methods

We have provided more detail about the methods section as outlined above.

Page 5, More detail has been added about how we purposively sampled the participants using a questionnaire collecting data of demographic and socioeconomic information.

Page 5, Details about the authors and interviewers have been move to the methods section and the wording for qualifications amended as suggested.

We have amended the sentence with analysis thematic analysis.

Results

Page 7, We have added more information about the sample.

Some of the themes have been changed to be more lay friendly and have shared the model with nurses and health professionals.

Page 8 (was 6), we have removed the text of “explanatory subthemes to explain”.

Page 8 (was 7), the quote has been changed to better make the point about the emotional impact of being a caregiver and the uncertainty of living with heart failure. Further text about the caregiver has been added.

Page 11 (was 9), we have added more text to highlight that the caregiver took part in the interview as it was an opportunity to talk about her experiences. We agree with the reviewer that this finding should be highlighted. We have added this to the discussion section and that another caregiver had appreciated the information about support services provided by the research patient information sheet helpful.

Page 11, the sentence implying a causal effect between caregiver anguish and social support has been modified.

Page 12 (was 10), the theme has been developed further with examples of when caregivers delayed attention to their own health. We added in the discussion as recommended that caregivers used the interview as a therapeutic effect (page 16).

There is now consistency with the subheadings with all those related to the themes now in italics.

Spelling of focus - throughout the paper we have now adopted the usually preferred version of one.

We are grateful for the final comments and agree that the terminology used was not user friendly. We have adapted the model and renamed the relevant subthemes.

Discussion

Pages 14 and 15, we have added as recommended more detail and references to the wider literature.

In particular we have included as suggested the work of Bury, Charmaz and Corbin and Strauss.

Page 17, we have provided text to support health professionals using the model in working with caregivers. We have also added a table of suggested questions to stimulate discussion with the caregiver about how they are really feeling – to get behind the smile.

Thank you again for the opportunity to review our manuscript and we look forward to the final decision.

VERSION 2 – REVIEW

REVIEWER	Kasey Boehmer Mayo Clinic, USA
REVIEW RETURNED	17-Feb-2017

GENERAL COMMENTS	The authors have addressed all of my concerns stated in the previous review, and I find the new version of the manuscript to be clear and acceptable for publication.
---